# MiRNAs and snoRNAs in Bone Metastasis: Functional Roles and Clinical Potential

**DOI:** 10.3390/cancers15010242

**Published:** 2022-12-30

**Authors:** Margherita Puppo, Mariam Jaafar, Jean-Jacques Diaz, Virginie Marcel, Philippe Clézardin

**Affiliations:** 1Université Claude Bernard Lyon 1, UCBL, 43 Bd du 11 Novembre 1918, 69100 Villeurbanne, France; 2INSERM, UMR_S1033, LyOS, Faculty of Medicine Lyon-Est, 7 Rue Guillaume Paradin, 69372 Lyon, France; 3DevWeCan Labex Laboratory, 69373 Lyon, France; 4INSERM, U1052, CNRS UMR5286 Centre de Recherche en Cancérologie de Lyon, 69000 Lyon, France; 5Centre Léon Bérard, 69008 Lyon, France; 6Institut Convergence PLAsCAN, 69373 Lyon, France; 7Department of Oncology and Metabolism, Medical School, University of Sheffield, Beech Hill Rd, Broomhall, Sheffield S10 2RX, UK

**Keywords:** non-coding RNA, post-transcriptional gene regulation, translation, rRNA chemical modifications, pre-metastatic niche, vicious cycle, osteoclast, osteoblast, extracellular vesicle, circulating biomarker

## Abstract

**Simple Summary:**

MicroRNAs and snoRNAs are regulators of gene expression in cells. In this review, we discuss the role of these two classes of small non-coding RNAs during the metastatic progression of cancers in bone. In primary cancer cells, microRNA and snoRNA expression is often dysregulated, leading to the acquisition of cell metastatic properties. Moreover, both microRNAs and snoRNAs can be released from cells, acting as intercellular mediators. MicroRNAs produced by primary cancer cells can remotely modulate the function of resident bone cells (osteoclasts and osteoblasts) to prepare the soil for cancer cell engraftment, a process called ‘pre-metastatic niche formation’. Then, microRNAs contribute to the creation of a positive feedback loop, the ‘vicious cycle’, between cancer and bone resident cells. We also present some evidence suggesting that snoRNAs might also be involved in these processes. Furthermore, we discuss the possibility that, in the future, microRNAs and snoRNAs may be used as biomarkers and/or therapeutic targets in the clinic.

**Abstract:**

Bone is a frequent site of metastasis. Bone metastasis is associated with a short-term prognosis in cancer patients, and current treatments aim to slow its growth, but are rarely curative. Thus, revealing molecular mechanisms that explain why metastatic cells are attracted to the bone micro-environment, and how they successfully settle in the bone marrow—taking advantage over bone resident cells—and grow into macro-metastasis, is essential to propose new therapeutic approaches. MicroRNAs and snoRNAs are two classes of small non-coding RNAs that post-transcriptionally regulate gene expression. Recently, microRNAs and snoRNAs have been pointed out as important players in bone metastasis by (i) preparing the pre-metastatic niche, directly and indirectly affecting the activities of osteoclasts and osteoblasts, (ii) promoting metastatic properties within cancer cells, and (iii) acting as mediators within cells to support cancer cell growth in bone. This review aims to highlight the importance of microRNAs and snoRNAs in metastasis, specifically in bone, and how their roles can be linked together. We then discuss how microRNAs and snoRNAs are secreted by cancer cells and be found as extracellular vesicle cargo. Finally, we provide evidence of how microRNAs and snoRNAs can be potential therapeutic targets, at least in pre-clinical settings, and how their detection in liquid biopsies can be a useful diagnostic and/or prognostic biomarker to predict the risk of relapse in cancer patients.

## 1. Introduction

Metastasis is the most advanced stage in cancer progression. Conventionally, metastasis starts when cancer cells that reside in a defined site (primary site) escape from it to colonise one, or more, distant sites (secondary sites), forming micro- and then macro-metastases [1]. However, recent findings suggest that the metastatic cascade starts even before the physical translocation of cancer cells from the primary site, with primary cancer cells producing a number of factors that allow distant sites to be ‘prepared’ to host disseminating cancer cells [2,3]. Among several organs (such as liver, brain, lymph nodes, etc.) that can be targets of disseminating cancer cells, the bone marrow is often a fertile soil for disseminated tumour cells, as demonstrated by the high incidence of bone metastasis in patients with cancer, mainly from primary breast, lung, prostate, kidney, melanoma, ovarian, and thyroid cancers [4]. In bone, two main resident cells, osteoclasts and osteoblasts, are important interplayers to allow (and further sustain) disseminated cancer cells to interact with this new microenvironment [2].

In cells, 99% of the total RNA content consists of non-coding RNAs (ncRNAs), with validated ncRNAs increasing every year [5]. As suggested by their name, ncRNAs are RNAs with no described potential to be translated into proteins, yet have a fundamental role in the regulation of gene expression [6]. In various diseases, including cancer, ncRNA expression, and thus its downstream pathways, are often dysregulated [7]. Not only do ncRNAs have an important role in bone metastasis progression [6], but they are also one of the keys to better understanding the molecular driving mechanisms of this metastatic disease. This review is focused on two classes of ncRNAs, microRNAs (miRNAs) and small nucleolar RNAs (snoRNAs), and their involvement in cancer progression towards metastasis, particularly in bone. While miRNAs have been largely investigated in the context of bone metastasis [8], snoRNAs have been poorly studied. However, there is evidence that snoRNAs can give rise to a specific class of miRNAs (called sno-miRNAs), thus sharing features with miRNAs. Moreover, snoRNAs also regulate bone homeostasis, as well as metastatic progression. Thus, we first summarise our current knowledge of bone metastasis, and we then illustrate the biogenesis of miRNAs and snoRNAs in cells. Next, we describe how circulating miRNAs and snoRNAs—especially the ones embedded in extracellular vesicles (EVs)—can contribute to the formation of a pre-metastatic niche in bone and metastasis progression, and how miRNA and snoRNA expressions in primary cancer cells can drive bone metastasis. Finally, we discuss the potential use and limitations of miRNAs and snoRNAs as prognostic biomarkers of cancer progression and anti-cancer therapeutic targets.

## 2. Current Understanding of Bone Metastasis

Bone metastasis happens when primary cells from different organs, which escape from their original site, find bone as an ideal place to seed and proliferate as a second cancer. The relative incidence of bone metastasis is evaluated based on the type of cancer, estimated to be 65–75% for breast and prostate cancers, 60% for thyroid cancer, 30–40% for lung cancer, 40% for bladder cancer, 20–25% for renal cell carcinoma, and 14–45% for melanoma [9]. Bone metastasis can affect bone in different ways: by (1) provoking an excess of bone formation (osteoblastic lesions), or (2) promoting bone destruction (osteolytic lesions), or (3) a mix of the two effects (mixed lesions) [8]. These defects result from the disruption of the finely-maintained equilibrium between bone formation and bone destruction, due to changes in upstream molecular pathways. For example, prostate cancer usually promotes osteoblastic lesions, while breast cancer bone metastasis is often characterised by osteolytic lesions [2]. As a consequence of disrupted bone homeostasis, bone metastasis is often associated with skeletal complications, which include bone pain, pathologic fractures, hypercalcemia, and spinal cord compression [10]. To date, bone metastasis is an incurable disease, with the exception of a few rare cases, and current treatments are only palliative and are mainly aimed at prolonging the patient’s life-span for a few years and reducing pain [11].

In bone, disseminated tumour cells will need to face a new microenvironment and interact with bone resident cells, such as osteoclasts and osteoblasts. Osteoclasts are monocyte-derived, multinucleated cells able to degrade the bone matrix (bone resorption) [2]. Osteoclast differentiation (osteoclastogenesis) is promoted by several factors that are physiologically expressed and released by osteoblasts or other bone resident cells, such as the receptor activator of nuclear factor kappa-beta ligand (RANKL), the macrophage-colony stimulating factor (M-CSF), and various cytokines [12,13]. The second main bone resident cells, osteoblasts, are derived from bone marrow mesenchymal stem cells (BMSCs). Mature osteoblasts secrete proteins that contribute to the production of the bone extracellular matrix that subsequently mineralises (bone formation). Osteoblast differentiation (osteoblastogenesis) is sustained by local factors, such as the transforming growth factor-beta (TGF-β), bone morphogenic proteins (BMPs), and the activation of the Wingless-INT (Wnt) pathway [14,15]. Both osteoclasts and osteoblasts contribute to the bone homeostasis to maintain healthy and strong bones. In metastasis, disseminated tumour cells can take advantage of the disequilibrium between osteoclast/osteoblast activity. For instance, breast cancer cells can take advantage of bone areas with high osteoclastic activity since, as bone is resorbed, growth factors released from the bone matrix can sustain cancer cell proliferation. Alongside this, disseminated tumour cells also find osteoblast-enriched niches, near the endosteum, which are very attractive, since they are highly vascularised and rich in factors that can promote cancer cell growth [16]. However, after the physical translocation of cancer cells in bone, it is well known that these cells alter osteoclast and osteoblast functions, further sustaining metastasis progression at the bone site [12,13,17]. The establishment of a positive feedback loop between cancer and bone resident cell metabolism is also described as a ‘vicious cycle’ [2]. As a cycle, it is difficult to identify a clear starting point and undoubtedly state if the first distortion at bone site is due to the activity of disseminated cancer cells or the bone remodelling itself, which favours the optimisation of a perfect niche. Moreover, the role of other bone resident cells, besides osteoclasts and osteoblasts, is extremely relevant in the progression of bone metastasis. As an example, it has been shown that osteocytes, which are actually the most abundant cell type (~95%), and are physiologically involved in bone remodelling in response to environmental and mechanical signals and stimuli, contribute to bone metastasis [18]. Moreover, as for other types of metastasis, immune cells [19] and cancer-associated fibroblasts [20] are important cellular players in the physio-pathology happening in bone and bone metastasis progression. A better understanding of molecular mechanisms (as well as driving players) behind the colonisation of disseminated cancer cells in bone is of central importance in the pre-clinical research.

Another interesting factor to consider is that, although bone metastasis occurs in more than 1.5 million cancer patients worldwide [21], the number of primary cancer cells actually able to circulate is estimated to be only the 0.02% of the total cell population [22]. This evidence raises the legitimate question: how can this apparently inefficient process be responsible for such a high recurrence of bone metastasis? Nowadays, it is widely accepted that primary tumours can favour the engraftment of (rare) disseminated cancer cells through a process known as ‘pre-metastatic niche formation’ (Figure 1). This concept is based on the evidence that secondary sites can naturally host, or be forced to host, future disseminating cancer cells [3], and this also implies that the choice of the secondary organ for metastasis is not a casual event but driven by specific events. For example, it has been demonstrated that primary tumour-secreted factors, such as the vascular endothelial growth factor A (VEGF-A) and placental growth factor (PGF), can mobilise hematopoietic cells in lungs before the arrival of metastatic cancer cells [23]. Interleukin-1β (IL-1β) and hypoxia-induced lysyl oxidase (LOX) are two other examples of primary tumour-derived soluble factors able to promote the colonisation of cancer cells in models of breast cancer bone metastasis [24,25]. The pre-clinical and clinical evidence of the existence of pre-metastatic niches is fast-expanding and makes the metastatic process even more complex. It is therefore essential to better understand the molecular mechanisms governing every step of metastasis in bone, including very early events, to propose new preventive and therapeutic options for cancer patients. In this respect, novel molecular mechanisms and potential innovative therapies might come from the emerging role of ncRNAs in promoting metastasis progression to the bone.

## 3. MiRNAs and SnoRNAs: Small yet Important Non-Coding RNAs

Currently, ncRNAs are defined by their length—small ncRNAs (sncRNAs) of 18–200 nucleotides, and long ncRNAs (lncRNAs) of >200 nucleotides—shape, and mechanisms of action; although, a certain difficulty in distinguishing categories exists due to the crossover of properties for some ncRNAs [6]. Among sncRNAs, microRNAs (miRNAs), short-interfering RNAs (siRNAs), PIWI-interacting RNAs (piRNAs), small rRNAs, transfer RNAs (tRNAs), small nucleolar RNAs (snoRNAs), and small nuclear RNAs (snRNAs) have been described [8]. SncRNAs regulate gene expression through various biological mechanisms that involve the interference, modification, and alternative splicing of other RNAs, often operated in conjunction with partner molecules, forming effective complexes. MiRNAs are the most studied sncRNAs, acting as promoters or repressors of cancer progression [26]. However, besides miRNAs, there are other classes of sncRNAs whose contributions to gene regulation are less known but important to understand. In this regard, snoRNAs consist of a class of sncRNAs mostly present in the nucleolus, a spatially defined compartment of the nucleus dedicated to the ribosome biogenesis [27]. Here, snoRNAs mainly target co- and post-transcriptional modifications of ribosomal RNAs (rRNAs), thus directly contributing to ribosome biogenesis [28]. As for miRNAs, snoRNA expression is associated with different stages of cancer progression, including metastasis [29]. Even if the biological functions of miRNAs and snoRNAs are very different, interestingly, small RNAs derived from snoRNAs (sdRNAs) that act as microRNAs exist, which are thus called sno-miRNAs [30]. Here, we summarise miRNA and snoRNAs biogenesis and functions.

### 3.1. Biogenesis and Biological Functions of MiRNAs

MiRNAs are 21–25 nucleotide-long sncRNAs that are highly conserved in animals and plants. MiRNA biogenesis in cells is a multi-step process that requires the consecutive action of two main enzyme complexes, the ribonuclease III Drosha/DGCR8 complex in the nucleus and the endonuclease Dicer/TRBP in the cytoplasm, to obtain the mature miRNA, which harbours biological functions (Figure 2) [6,31]. From each miRNA precursor, two independent single-strand RNAs originate, the 5′–3′ and 3′–5′ filaments. Both filaments correspond to mature miRNAs that can potentially interact with an AGO protein and the RNA-induced silencing complex to form an effector complex. The effector complex uses the incorporated mature miRNA as a template to recognise RNA targets, usually messenger RNAs (mRNAs), based on the complementarity of sequences [6,31]. When there is 100% complementarity between miRNA:mRNA sequences, this leads to the degradation of the mRNA target by the recruitment of various complexes that lead to mRNA deadenylation, decapping, and 5′–3′ cleavage. In contrast, when the interaction between miRNA and its mRNA target is limited to the ‘seed sequence’ of the mature miRNA (i.e., the first 2–8 nucleotides in the 5′–3′ direction), it has been shown that the translation of the mRNA target is inhibited [6,31,32]. Although most miRNAs are reported to repress the expression of their targets by these two mechanisms, some miRNAs have been shown to upregulate the translation of their mRNA targets by directly acting in trans, or indirectly by abrogating the actions of repressive effector complexes [6,31,33]. Interestingly, there is evidence of how translational inhibition and transcript destabilisation are tightly interconnected, supported by the fact that mRNA targets, under miRNA repression, are associated with actively translating ribosomes [34]. Whether translation is required for miRNA repression of mRNA targets is still a matter of debate [34]. It needs to be considered that these mechanisms are often cell-specific, and they require specific conditions (e.g., gene mutations, external stimuli), suggesting that studying novel physio-pathological conditions would improve our understanding of miRNA mechanisms of action.

The extremely wide regulatory capacity of miRNAs stands on the potential of a single miRNA to bind, and thus regulate, several targets. Thus, a dysregulation of the expression of a single miRNA is sufficient to profoundly reprogram gene expression and shape a specific network of genes encoding proteins involved in diverse biological pathways, allowing extensive changes in cell identity and/or behaviour. Thus, miRNA expression and activity are finely regulated in cells, including by their mRNA target themselves. In fact, the up-regulation of mRNA targets can result in the mRNA sponging miRNAs—and for this reason, they have also been called ‘miRNA sponges’—thus preventing the latter from regulating other transcripts. Additionally, other ncRNAs, such as lncRNAs, can act as miRNA sponges, preventing miRNA repression of their targets [35]. Thus, there is a delicate equilibrium between the expression of miRNAs and their RNA targets, which can also be easily disrupted. Abnormal miRNA expressions in cancer cells are common, and so far, both oncomiRs and oncosuppressor miRNAs have been identified in cancer. However, we only partially understand the impact of miRNAs in cancer progression and how to translate our knowledge into innovative therapeutic options for cancer patients.

### 3.2. Biogenesis and Biological Functions of SnoRNAs

Another class of sncRNAs corresponds to snoRNAs, which are 60–300 nucleotide-long RNA mainly localised in the nucleolus of cells [36,37]. SnoRNA functions mostly consist in guiding the addition of chemical modifications to ribosomal RNAs (rRNAs) and, to a lesser extent, regulating the processing of rRNAs [27,38]. In addition, they are also involved in chemically modifying other RNAs, including transfer RNAs (tRNAs) and mRNAs [39]. Conventionally, snoRNAs are divided into two main classes based on their biological functions and structure: box C/D (snoRDs) and box H/ACA (snoRAs) snoRNAs, responsible for the 2′O-methylation of ribose (2′Ome) and pseudouridylation (Ψ), respectively. Moreover, while snoRDs are typically 60 to 90 nucleotides long, snoRAs are larger snoRNAs, which range from 120 to 140 nucleotides. These snoRNAs specify the location of RNA chemical modifications (Figure 2). A third snoRNA subfamily, the small Cajal bodies RNAs (scaRNAs), localised in the Cajal bodies of the nucleus, are mainly involved in aiding the formation and maturation of spliceosomes and ribosomes, besides other functions in common with other snoRNA classes [40].

SnoRNAs are characterised by the presence of specific structures (k-turn in snoRDs; stem-loop in snoRAs), as well as two sets of conserved sequence motifs that flank the complementary sequence of the RNA target: the ‘box C’ (5′-RUGAUGA-3′, where R is a purine) near the 5′ end, and the ‘box D’ (5′-CUGA-3′) near the 3′ end in snoRDs; and the ‘hinge box’ (H box, 5′-ANANNA-3′), and the ‘ACA Box’ (5′-ACA-3′) at the 3′ end of snoRAs. Their canonical activity of RNA modification relies on their ability to associate with proteins in order to form a stable small nucleolar ribonucleoprotein (snoRNP). The box C/D snoRNP consists of three structuring core proteins (Nop56, Nop58, 15.5k/NHP2L1) and the methyltransferase fibrillarin (FBL). The mature box C/D snoRNP interacts in a sequence-dependent manner with the rRNA through the snoRD, which guides FBL on the fifth nucleotide from the 5′-end D box, thus specifying the precise location for 2′Ome. Similarly to box C/D snoRNAs, snoRAs associate with four proteins and form H/ACA stable and functional snoRNPs, including the three core proteins Nhp2, Nop10 and Gar1, as well as with the pseudouridine synthase dyskerin (DKC1), which is guided on the uridine of interest through snoRA:rRNA interaction [38]. 

In humans, most snoRNAs mature from the introns of both non-coding and coding genes mainly related to ribosome biogenesis. The biogenesis of most intronic snoRNAs includes co-transcription of the host gene, splicing, debranching of the intron lariat, and exonucleolytic digestion in the nucleoplasm. Mature snoRNPs are transported to the nucleolus, where they can exert their canonical roles [41]. In some cases, both snoRDs and snoRAs can be further processed into smaller RNAs, known as miRNA-like snoRNA-derived miRNAs (sdRNAs, sno-miRs). SdRNAs have been reported in several organisms, and they have been shown to play a role in the key biological features of metastatic progression. The main function of snoRNAs consists in the 2′Ome and Ψ of rRNA. During the last decade, several studies reported that alterations of rRNA chemical modifications occur in different pathophysiological contexts, including cancer, to modulate intrinsic activities of the ribosome, thus contributing to translational reprogramming to specify particular phenotypes [27]. For instance, alteration of the rRNA 2′Ome pattern regulates the translation of a subset of mRNAs containing a specific cis-regulatory element, the internal ribosome entry site (IRES) [42,43]. This emerging concept, placing the ribosome at the heart of translational regulation, shaded into light the importance of snoRNAs in human biology. In addition, it has recently emerged that snoRNAs could have other non-canonical functions by contributing to additional types of rRNA chemical modifications (e.g., acetylation), or by regulating mRNA maturation (e.g., splicing, 3′-end processing, exosome recruitment) through either RNA or protein interactions. Overall, the emerging role of snoRNAs in regulating gene expression both at post-transcriptional, notably translational, and transcriptional levels, raises novel opportunities in understanding physio-pathological processes. This is particularly true in cancer, where snoRNAs display oncogenic and suppressive activities, and appear as useful tumoral and circulating biomarkers [44,45,46,47,48]. However, we also acknowledge the lack of experimental evidence regarding snoRNA role in bone metastasis progression, which we highlighted later on in this review.

## 4. Role of Circulating MiRNAs and SnoRNAs in Bone Metastasis

A common ground between miRNAs and snoRNAs is the fact that both can circulate in biological fluids, such as blood and lymph [49]. Compared to other RNAs, their small size as well as their interaction with core proteins protect them from a massive degradation both as free and embedded forms, making sncRNAs stable, secreted, circulating molecules. As an embedded form, they are usually within extracellular vesicles (EVs) that derive from cells. The term EVs describes a heterogeneous class of vesicle organelles that originate from cells, and that are mainly categorised based on their size and mechanism of action in exosomes, microvesicles, or apoptotic bodies [50]. EVs carry biological material (proteins, lipids, DNA, RNA) that usually reflects what is produced by parental cells. Interestingly, one of the major components of the EV cargo are ncRNAs, such as miRNAs (about 26%) and to a lesser extent snoRNAs (0.6%). The EVAtlas database on EV ncRNA-content is available [51]. Recently, the evidence that tumour cell-derived, EV-encapsulated miRNAs, mainly, and snoRNAs promote tumorigenic processes has gained more interest in pre-clinical research [52]. We will discuss below the role of both free and EV-associated circulating miRNAs and snoRNAs in the metastatic niche formation (Figure 3), and the vicious cycle (Figure 4) in bone.

### 4.1. MiRNA and SnoRNA Roles in the Formation of a Pre-Metastatic Niche

As previously discussed, the bone marrow stroma, which is enriched with cytokines and growth factors, is an advantageous environment for the homing and outgrowth of metastatic cells [53]. Besides the intrinsic characteristics of bone, as an attractive and fertile metastatic site, there is evidence that bone niches can be further created or promoted by the remote action of primary tumour cells [54]. Pre-clinical and clinical research have provided evidence that EVs, as systemic factors, could create ideal conditions at distant sites, allowing disseminated cancer cells to colonise bone marrow. Thus, primary cancer cells can remotely ‘educate’ distant sites in bone, by secreting EVs, to further receive and host disseminated cancer cells even before the metastatic process starts at the primary site. Indeed, it has been demonstrated that EVs can be internalised by resident cells, such as osteoclasts and osteoblasts, and modulate their maturation and activity [3]. For example, the internalisation of EVs produced by multiple myeloma cells can promote the differentiation of osteoclasts [55], while prostate cancer cell-derived EVs inhibit it [56]. Moreover, the pre-treatment of animals with tumour cell-derived exosomes increases metastatic burden in murine models of prostate cancer, making the bone marrow the preferential target for these tumour cells [57].

Some studies aiming to characterise the EV content also explored the role of ncRNAs as molecular players of changes induced by EVs. Up to now, EV-derived miRNAs are the best studied ncRNAs from the exosomal cargo, and several of them have been shown to play a direct role in cancer progression. First, the fundamental role of the EV-miRNA content has been proven by a study showing that, while wild-type EVs produced by prostate cancer cells contribute to creating a pre-metastatic niche in bone, EVs produced by Dicer-depleted tumour cells have less effects on bone cells [58]. Then, a direct effect of specific EV-cargo miRNAs on the preparation of pre-metastatic niche was explored in various studies (Figure 3). For example, breast cancer-secreted miR-105 can be delivered, embedded in EVs, to endothelial cells and can promote tumour metastasis [59]. Although this work mainly focuses on lung and brain metastases, these findings might be extended also to bone. Slightly more controversial, there is evidence that the well-known oncogene suppressor miRNA miR-200, produced by primary breast cancer cells and then encapsulated in EVs, promotes metastasis [60]. Another study on breast cancer showed that tumour-derived, exosome-delivered miR-20a-5p facilitates osteoclastogenesis by targeting SRCIN 1, previously known for its role in cancer progression [61]. A similar study has been conducted in lung cancer, where the exosomal miR-214 can be released by both lung cancer cells and osteoclasts, mutually and positively contributing to osteoclast activation in bone, thereby favouring formation of osteolytic metastases [62]. In prostate cancer, tumour-derived miR-141-3p embedded in EVs can be taken up by osteoblasts, promoting their activity, and indirectly compromising the function of osteoclasts to ultimately promote the formation of osteoblastic bone metastases [63]. Moreover, EVs-embedded miRNAs can have synergic effects with secreted proteins. For instance, EV-embedded miR-19a together with the integrin-binding sialoprotein (IBSP) are secreted by breast cancer cells, and they synergistically influence the bone microenvironment [64]. While IBSP creates an osteoclast-enriched niche, exosomal miR-19a induces osteoclastogenesis, two factors that contribute to creating a favourable site for breast cancer metastasis [64]. Another study identified exosomal miR-940 as being highly expressed in prostate cancer cells, which usually induces an osteoblastic phenotype in the bone metastatic microenvironment [65]. Interestingly, the artificial expression of miR-940 in breast cancer cells, which usually produce osteolytic bone metastasis lesions, induces extensive osteoblastic lesions in animals by promoting osteoblast maturation [65]. This study is particularly interesting as it demonstrates how it is possible to reprogram the bone metastatic microenvironment through the secretion of a single miRNA, in this case miR-940, by modulating osteoblast activity. This clearly shows how tumour-derived miRNAs are powerful regulators of gene expression, and how important is to track down these modulations for a more comprehensive understanding of metastasis.

The role of EVs-embedded snoRNAs in the formation of a pre-metastatic niche in bone has never been investigated. However, the development of a RNA-seq approach, including the thermostable group II intron reverse transcriptase sequencing (TGIRT-seq), dedicated to small structured ncRNA and based on the use of group II intron-encoded RTs instead of low-fidelity retroviral RTs, demonstrated that snoRNAs can be detected and identified in EVs [52,66]. Such an approach led to surprising clinical observations, suggesting that snoRNAs may serve as novel peripheral blood plasma-EV-derived biomarkers for monitoring astronauts’ health [67]. Indeed, this study revealed that several snoRNAs, including SNORA74A, were significantly dysregulated in peripheral blood plasma EVs from astronauts 3 days after the shuttle missions, compared to 10 days before the missions. Whether snoRNA-associated EV might be involved in metastatic progression clearly needs to be addressed. Overall, the lack of knowledge on the role of snoRNA in premetastatic niche formation suggests that a more comprehensive study aiming to explore the secretome from primary cancers—and the potential role of circulating small RNAs, besides miRNAs, in the remote control of distant organs—is a very attractive opportunity to better understand bone metastasis mechanisms.

### 4.2. MiRNA and SnoRNA Roles in the Vicious Cycle in Bone

Since both osteoclasts and osteoblasts are the main regulators of bone homeostasis, they play an important role in allowing the seeding and sustaining outgrowth of metastatic cells [2,4]. Thus, EV-derived cargo that can affect any of the bone resident cells may be responsible for not only creating the pre-metastatic niche, but also promoting a series of events that establish a complex crosstalk between bone resident and metastatic cells. This concept is known as the vicious cycle (Figure 1). While miRNAs have clearly been shown to act as cell mediators in this vicious cycle, the role of snoRNAs remains largely unknown (Figure 4).

Regarding miRNAs, several examples can be cited. MiR-214, which has been found to be highly expressed in lung adenocarcinoma, is also shown to mediate intercellular communication between osteoclasts and osteoblasts [62]. In this study, exosomal tumour-derived miR-214 is proposed as further intercellular mediator between the primary tumour and osteoclasts. Specifically, tumour-exosomal miR-214 stimulates osteoclast differentiation, consequently increasing bone resorption, and the availability of cytokines and growth factors in the bone environment. Moreover, miR-214 can be secreted from osteoclasts. Thus, targeting miR-214 might be a good strategy to interrupt the vicious cycle at the bone metastatic site [62]. In breast cancer bone metastasis, exosomal miR-19a and the integrin-binding sialoprotein (IBSP), derived from oestrogen-receptor positive breast cancer cells, induce osteoclastogenesis and create a bone microenvironment enriched with mature osteoclasts [64], which is known to attract metastatic cancer cells. In prostate cancer, EVs that derive from prostate cancer cells increase osteoblastic activity and metabolism and impair bone resorption [68]. In this study, miR-26a-5p, miR-27a-3p, and miR-30e-5p have been identified as the abundant cargo of these EVs. Moreover, these miRNAs are involved in the suppression of the BMP-2-induced osteogenesis, suggesting a role in the suppression of bone resorption [68], and pointing out the importance of miRNAs as EV-cargo effectors. Another study identified miR-92a-1-5p as an abundant miRNA in exosomes from prostate cancer cells that directly target collagenase 1-A1 (Col1A1), promoting osteoclast differentiation and inhibiting osteoblastogenesis [69], which is quite a surprising finding knowing that prostate cancer metastases in bone have usually an osteoblastic phenotype. However, this illustrates how bone remodelling can be remotely modulated by miRNAs to allow the future hosting of cancer cells.

Although not precisely studied in the context of the vicious cycle in bone, the relationships between bone homeostasis and snoRNAs have been identified in different pathologies. First, in vitro treatment of primary osteoclasts with an anti-HIV drug (Tenofovir), which promotes loss of bone mineral density, significantly reduced SNORD32A expression, although its role in osteoclast dysfunction remains to be determined [70]. Second, SNORD116 loss in a mouse model of the Prader–Willi familial syndrome is sufficient to reduce both bone mineral content and density, by reducing osteoblast differentiation without alteration in osteoclastogenesis [71,72]. Moreover, SNORD116 along with other snoRNAs have been found dysregulated in serums and tissues of mice affected by osteoarthritis and joint ageing, further suggesting the potential use of snoRNAs as biomarkers [73]. Alteration in snoRNA pattern has also been observed in senescent BMSCs, which are capable of self-renewal into different cell types including osteoblasts, reinforcing the association between snoRNAs and physiological bone formation. In cancer, it has been shown that snoRNAs contribute to the metastatic potential of p53-induced osteosarcoma [74]. Indeed, the deletion of the transcription factor ETS2 in conditional osteoblast mutant p53 mice reduces the expression of a panel of 24 snoRNAs and reverses the metastatic phenotype of mutant p53 without affecting osteosarcoma development. Overall, these data support the notion that snoRNAs contribute to osteoclast/osteoblast balance in a physiological context and/or its imbalance in different diseases, including cancer. However, the role of snoRNAs in the vicious cycle in bone has not been investigated yet.

Of particular interest is the recent discovery that the tropism of tumour-derived exosomes can be site-specific due to the expression of specific proteins at the exosome surface [75]. In bone, it has been identified that L-plastin, an actin-binding protein, as a component of exosomes from breast cancer cells, is able to activate osteoclasts [76]. The same study demonstrated that peroxiredoxin-4 (PRDX-4) is also implicated in this process, and that higher levels of both L-plastin and PRDX-4 are associated with a higher risk to develop bone metastasis in breast cancer patients [76]. MiRNAs and snoRNAs, being cargo, cannot directly drive the tropism of exosomes, which happens thanks to protein–protein recognition. However, they can regulate the expression of proteins that can be expressed at the exosome surface [77], suggesting a regulatory role for miRNAs and snoRNAs in the tropism of EVs.

## 5. MiRNA and SnoRNA Expression in Cancer Cells and Their Roles in Bone Metastasis Progression 

After, or besides, accomplishing the preparation of a pre-metastatic niche, both miRNAs and snoRNAs expressed in cancer cells can actively promote bone metastasis. Within cancer cells, both miRNAs and snoRNAs can be regulators of the expression of (i) oncogenes, (ii) tumour suppressor genes, and/or (iii) genes involved in the acquisition of metastatic properties to promote EMT/MET and cancer stemness. 

Several well-known miRNAs have been shown to promote metastasis progression in distant organs, including in bone. MiR-10b has been the first identified miRNA associated with breast cancer metastasis, which also promotes the early stage of bone metastasis formation, but is not involved in the formation of primary breast tumours [78], making miR-10b a perfect example of the miRNA involvement in metastasis. In breast cancer, a number of miRNAs have been associated with the acquisition of metastatic properties (miR-1976, miR-429, miR-30 family, miR-205, miR-143, miR-20a-5p, miR-34a-5p, miR-203, miR-135), osteomimicry (miR-218, miR-135, miR-203), and disruption of the crosstalk between tumour cells and the bone microenvironment; these different properties have recently been extensively discussed in a review article [79]. Similar roles for miRNAs have also been reported in cancers that are prone to colonise bone, such as prostate [80], lung [81], kidney [82], melanoma [83], ovarian [84], and thyroid [85] cancers. 

The deregulation of some snoRNAs has been shown to promote acquisition of migratory, invasive, and stemness capabilities both in vitro and in vivo (Figure 5). For instance, in mouse models, overexpression of SNORD38 in primary lung cancer has been shown to increase distant metastases [86], while SNORA23 overexpression in pancreatic PDAC cell lines promotes liver metastases in animal models [87]. Meanwhile, although it is widely recognised that snoRNAs have a strong impact in cancer biology, most molecular mechanisms involving snoRNAs need to be identified. At present, no study ever investigated the role of snoRNAs in promoting bone metastasis. However, most of the in vitro studies were performed using primary tumours with a bone tropism, suggesting a role of snoRNAs in metastatic progression in bone. Indeed, experimental modulation of snoRNAs has been shown to regulate migration/invasion in different cancer cell lines derived from prostate (SNORA42, SNORA55) [88,89], breast (SNORA7B, SNORA71A, SNORD50A/B) [90,91,92], lung (SNORA42, SNORA47, SNORA71A, SNORD38, SNORD78) [86,93,94,95,96], and ovarian cancer (SNORA70E, SNORA72, SNORD89) [97,98,99]. In addition, some of these snoRNAs have also been shown to modulate stemness capabilities in these different cancer types having a bone metastatic tropism (SNORD78, SNORA42, SNORD89, SNORA72, SNORA71A). For instance, snoRNA profiling in tumoral tissue allowed the identification of specific snoRNAs whose expression in primary tumours is associated with lymph node metastasis. In endometrial cancer, the box C/D snoRNA SNORD89 expression level is higher in patients with lymph node metastasis than those without lymph node involvement. Furthermore, SNORD89 overexpression promotes cell migration and inhibits BIM mRNA translation through higher activity of 2′Ome, thus dysregulating the Bim/Bcl2/Bax signalling pathway leading also to apoptosis inhibition [100]. In breast cancer, SNORA71A is highly expressed in metastatic breast cancer tissues compared to non-metastatic samples. It was shown that SNORA71A, through the regulation of ROCK2, promotes migration, invasion, and EMT in breast cancer cell lines. In fact, SNORA71A recruits the mRNA-stability-regulated protein G3BP1, which in turn binds and stabilises the ROCK2 mRNA [90]. Additional snoRNAs have been shown to regulate EMT, such as SNORA42 [88], SNORA71A [90,96], and SNORD78 [94]. Specifically, SNORA72 has been shown to regulate migration/invasion and stemness through the activation of the cMyc/Notch pathway [99], SNORA47 through the activation of the PI3K/AKT and MAPK/ERK pathways [95], SNORA55 through the TNF/GHRH pathways [89], and SNORD50A/B through p53 [91]. Although these signalling pathways are regulated in response to snoRNA modulation, molecular mechanisms behind them remain poorly described and mostly rely on non-canonical activities of snoRNAs. Another study reported SNORA70E as a promoter of cell migration/invasion by modulating the alternative splicing of PARPBP [97]. As stemness is a key feature in the metastatic progression, it is particularly interesting that a study identified a signature of 22 snoRNAs associated with an elevated activity of the aldehyde dehydrogenase (ALDH) enzyme, a marker of stemness, in tumour-initiating cells from non-small cell lung carcinomas [44,93], suggesting that these snoRNAs are good candidates to be investigated at a functional level. 

Evidence suggests a role for sdRNAs in metastasis. In breast cancer, the snoRNA-93 (HBII-336) is processed into a smaller RNA (sdRNA-93) that is markedly overexpressed in the metastatic MDA-MB-231 cell line as compared to MCF-7 cells, which are poorly metastatic [101]. Importantly, sdRNA-93 suppression in MDA-MB-231 cells decreases cell invasion, whereas sdRNA-93 enhancement increases tumour cell invasion. Mechanistically, several targets of sdRNA-93 have been identified, including Peroxisomal Sarcosine Oxidase (Pipox) whose expression is regulated by sdRNA-93.

Altogether, these data show that miRNAs and snoRNAs expression in primary tumour cells could have a direct impact on bone metastasis formation and progression. While this has been broadly studied and described with miRNAs, it still needs to be investigated for snoRNAs.

## 6. Circulating MiRNAs and SnoRNA as Potential Biomarkers

Since metastasis is responsible for 90% of cancer-associated mortality in patients [1], its early detection may reduce the death rate and improve overall survival of patients. Thus, pre-clinical research on new biomarkers able to predict metastasis recurrence is essential. In cell-free liquid biopsies, which are a minimally if not a non-invasive method to analyse body fluids from patients, a variety of molecules (such as DNA and RNA, including miRNAs and snoRNAs), either circulating as free or EV-embedded molecules, can be quantified and used as biomarkers of disease progression, such as in cancer. While plasma cell-free DNA is widely used in clinical practice [102,103], circulating RNA (and mainly miRNAs and snoRNAs) starts to be a promising new technology for future clinical use. Here, we present some pilot studies performed to identify potential RNA-based (miRNAs, snoRNAs) circulating biomarkers and we discuss how these findings can be translated into the clinic, highlighting limitations and strengths.

### 6.1. Use of Circulating MiRNAs and SnoRNAs as Biomarkers: Some Examples

So far, some miRNAs isolated from circulating EVs have been demonstrated to be able to mirror signatures of primary tumours. One example is the exosomal miR-373 reported to distinguish between breast cancer subtypes [104]. As in breast cancer, similar evidence is reported for lung cancer, with two miRNA signatures with clinical validation in large cohorts of patients [105], nasopharyngeal carcinoma, with a cluster of significant overexpressed miRNAs [106], and in ovarian cancer, with a cluster of 3 miRNAs from the miR-200 family being significantly associated with cancer recurrence and overall patient survival [107].

Regarding snoRNAs, some circulating EV-embedded snoRNAs have been associated with clinical outcome. Up to now, most of the studies compared cancer and healthy patients, providing evidence that circulating snoRNAs can be useful diagnostic biomarkers. For example, in hepatocellular carcinoma, four snoRNAs were upregulated (SNORD3A, SNORD91B, SNORD65, SNORD55) and two were downregulated (SNORD116-3/24) in the plasma of cancer patients compared to healthy donors [52,108]. Furthermore, the detection of free-circulating snoRNAs (e.g., SNORD33, SNORD76 and SNORD66) in body fluids, such as plasma, serum and sputum, show relevance in the diagnosis of various cancers, including lung and prostate cancers [44,89,109]. However, analysis of circulating snoRNAs, as putative biomarkers of metastatic progression, has not been conducted so far.

Nevertheless, several pieces of evidence support the notion that snoRNAs might be powerful biomarkers of metastatic progression. First, snoRNA expression in primary tumours has been associated with poor prognosis in numerous cancers. Specifically, SNORD44 (RNU44) is significantly associated with distant disease-free survival in breast cancer, suggesting that this snoRNA could be used as a biomarker of metastatic progression. Second, some studies reported differential expression levels in snoRNAs between primary and metastatic tissues, in particular when considering lymph node invasion. For instance, deep sequencing of patient-derived samples from normal prostate, and prostate cancer at different stages of the disease, revealed that the box C/D snoRNA SNORD78 expression and its derived sdRNA (sd78-3′) are highly expressed in patients that develop metastasis, especially lymph node metastases. Strikingly, the expression of sd78-3′ and its precursor SNORD78 were already significantly higher at the time of surgery, suggesting their early involvement in prostate cancer progression. However, more studies must be conducted in order to understand how SNORD78 and sd78-3′ are deregulated in prostate cancer, and if they act concomitantly in order to promote metastatic progression [110]. Finally, the study of Crea and colleagues [89] revealed that a single snoRNA, SNORA55, can be an interesting circulating biomarker for both diagnosis and prognosis. SNORA55 is not only present at high levels in the serum of prostate cancer patients compared to healthy donors, but also in primary tumour samples of metastatic patients compared to those from patients who did not relapse. It has to also be noted that SNORA55 upregulation displays significantly shorter relapse-free survival after surgery, which gives SNORA55 the distinctive feature of being a predictor of post-prostatectomy outcome.

Finally, as already stated, circulating EVs derived from tumour cells with a bone metastasis tropism are taken up by specific recipient cells in the bone marrow, enabling the formation of a pre-metastatic niche to attract tumour cells [75]. It has been demonstrated in mice that circulating snoRNAs are able to modulate 2′Ome in distant tissues [111]. However, an analysis of liquid biopsies from cancer patients with different metastatic status is lacking to demonstrate the usefulness of snoRNAs/miRNAs as circulating biomarkers to predict metastatic progression in distant organs, such as bone. 

### 6.2. Technical Strengths and Limitations of MiRNA and SnoRNA Biomarkers

Before being able to translate the sncRNA detection into the clinic, it is still important to consider the pros and cons in analysing serum-derived sncRNAs. The main asset of ncRNAs, as compared to other circulating biomarkers, remains their abundance and stability in biological liquids, particularly in serum, in comparison to other RNAs [112]. In serum, miRNAs, piwi-interacting RNAs, transfer RNAs, snoRNAs, small nuclear RNAs are indeed the core of circulating RNAs that have been detected by small RNA-seq in a large cohort of human serums, which took into consideration also lncRNA and mRNA fragments [113]. Moreover, miRNAs stability can last decades [114], allowing retrospective studies. Importantly, EVs that circulate in biological fluids protect miRNAs from degradation, making EVs-embedded miRNAs highly suitable for clinical detection. This, however, still needs to be proven for snoRNAs.

Some limitations should be also considered before transferring miRNAs/snoRNAs into the clinic. In fact, miRNAs that are detectable in the serum can be contaminated by platelet-derived miRNAs released during clot formation, which can influence downstream analyses [115]. Additionally, plasma can be affected by miRNA/snoRNAs contamination from blood cells, such as erythrocytes [116]. It is also important to take into consideration the fact that EV isolation mainly relies on ultracentrifugation methods, which do not exclude contamination by other types of vesicles. Other contaminants such as lipoproteins and serum-derived materials should also be taken into consideration when working with EVs [117]. Thus, a wise choice of the kinds of samples to use is very important. Improvements in the methodology and tools to collect cell-free miRNAs/snoRNAs need to be taken into consideration, and the purification of EVs could be a useful solution in order to enrich the circulating-miRNA/snoRNA fraction. 

Other limitations to ncRNA detection are due to their short length, their high structuration, and degree of homology. For snoRNAs, homology relies on the presence of conserved C/D or H/ACA motifs. For snoRNAs and miRNAs, homologies also come from the existence of several family members with common sequences. Overall, these limitations make detecting specific miRNAs and/or snoRNAs a real (technical) challenge. Likely, quantitative real-time PCR (RT-qPCR) is the most common technique used in the clinic [118]. The untargeted next-generation sequencing (NGS) for ncRNAs detection can be a second option; however, this method is more for discovery phases and pre-clinical settings, while in the clinic it could result in more overall expenses. In addition, it is important to keep in mind the fact that ncRNAs NGS technology implies additional technical steps, such as library purification on agarose gels for the snoRNA fraction, which could be difficult to use in routine. Another technique to consider could be the use of droplet digital PCR (ddPCR) for the absolute quantification of ncRNAs [119]. Finally, it has been reported that miRNA/snoRNA expression in humans can be affected by sex, ethnicity, age, lifestyles, and circadian rhythms [120,121,122]. This needs to be taken into consideration and more studies, with larger patient cohorts, should be performed to avoid bias. For snoRNAs, another limitation consists in the low detection levels in plasma or EVs (between 0.01% and 0.6%) compared to miRNAs (between 40% and 80%) [123]. Whether this difference resides in the fact that the detection methods are not rigorous enough—neither adapted for snoRNAs detection, or the fact that snoRNAs are not abundant in the circulation—is an important point to evaluate. 

To conclude, it needs to be considered that it is unlikely that the detection of one ncRNA will be able to give a proper readout in the clinic. The combination of several biomarkers, including a ncRNA-based signature, seems more appropriate. For instance, in sputum, while a panel of three miRNAs or of two snoRNAs gives high sensitivity/specificity to diagnose early lung cancer (AUC 0.90 and 0.86, respectively), the combined use of these five snRNAs considerably improves confidence in the diagnosis (AUC 0.94) [124]. Ideally, these ncRNA biomarkers alone, in combination, or in multispecies signature (protein, DNA, RNA) should be predictive of bone relapse from different cancers. 

## 7. Therapeutic Opportunities from MiRNAs and SnoRNA Use 

The use of miRNAs and snoRNAs as therapeutic targets is a promising opportunity that has the potential to radically change the clinical approach to cancer and its metastatic disease, particularly in bone metastasis, which is still incurable. 

Also, in this case, miRNAs have been largely investigated in comparison to snoRNAs. Since a single miRNA can target multiple pathways at the same time, a miRNA-based therapy could have an enhanced effect in comparison to more specific therapies targeting a single molecule; however this can also lead to more off-target effects. So far, two miRNA-based therapeutic strategies have been developed and investigated at the experimental level. The first strategy aims to deliver, to tumour cells, onco-suppressive miRNAs which are usually downregulated, in order to restore a ‘less aggressive’ cell phenotype. Synthetic miRNA-mimics—characterised by the same sequence of ‘real’ miRNAs—are the most studied molecules able to mimic the biological functions of miRNAs. In general, miRNA-mimics are well tolerated by cells, without any major cytotoxic effect in normal tissues [125]. However, other delivering agents have been used in pre-clinical settings, such as small molecules (hypomethylating agents) and vectors expressing specific miRNAs [125]. The second strategy implies the use of molecules able to link to miRNAs, in this specific case oncomiRs, thus limiting their biological availability within cancer cells to restore normal expression levels. A few miRNA-inhibiting molecules have been developed so far, including miRNA sponges, antisense anti-miR oligonucleotides, locked nucleic acid anti-miRNAs, antagomiRs, miRNA masks, and small molecule inhibitors of miRNAs [125,126]. Interestingly, some miRNAs that have been reported in pre-clinical studies might be good candidates for miRNA-based therapies. As an example, it has been shown that the administration of miR-10b antagomiRs in animal models can efficiently inhibit breast cancer lung metastasis without causing any adverse effects [26]. Similarly, miR-125b mimics have been reported to delay breast cancer bone metastasis progression in animals [127]. Moreover, for clinical application, the possibility to carry miRNA-mimics or antagonists into exosomes, liposomes, nanoparticles, or other vectors should be taken into consideration since it can increase delivery efficiency and reduce cell toxicity. For instance, miR-155 exosomal delivery to cells, hepatocytes, and macrophages in this specific study, seems to be a promising approach to target biomolecules [128], which can be expanded to the context of bone metastasis with the use of miRNAs and snoRNAs reported to have an effect in bone. However, off-target effects, for example to other organs, should be taken into consideration at a pre-clinical level.

Even if many miRNA-based therapies have been experimentally investigated at a preclinical level, only a few have entered early-phase clinical trials. One promising yet unsuccessful clinical trial (#NCT01829971, from https://clinicaltrials.gov/, accessed on 8 September 2022)—interrupted at phase-I due to five cases of adverse reactions—has been conducted, injecting lipid nanoparticles-encapsulated miR-34a mimics (MRX34) in 155 advanced-stage patients with refractory solid tumours. MiR-34a has been previously demonstrated to reduce CD44 protein levels at the tumour cell surface and reduce metastasis formation in animal models of prostate cancer [129]. Possibly, a different route of administration, or type of miRNA-mimic carrier, should be considered in the design of a future trial for miR-34. A second miRNA-mimic investigated in a phase-I trial (#NCT02369198, from https://clinicaltrials.gov/, accessed in September 2022) used miR-16 embedded in targeted minicells, which were injected in 27 patients with malignant pleural mesothelioma and non-small cell lung cancer, and no toxicity was demonstrated [130]. However, a phase-II trial with a larger number of patients is needed. No clinical trial has been conducted to evaluate the effect of miRNA-base therapies specifically in the context of bone metastasis.

Up until now, snoRNA-based therapies have not yet been investigated. However, based on the similarities between snoRNAs and miRNAs highlighted in this review, we can speculate a potential use of snoRNAs for targeting cancer metastasis. Interestingly, snoRNA expression can be modulated in cells by the systemic injection of antisense oligonucleotides (ASOs) in animal models [131], suggesting a potential use of ASOs as therapeutic agents. Therefore, it would be very interesting, in the near future, to better explore snoRNA and miRNA as therapeutic agents.

## 8. Conclusions and Future Perspectives

Here, we reported the role of two classes of small non-coding RNAs, miRNAs and snoRNAs, in bone metastasis. Even if more research on miRNAs has been conducted so far compared to snoRNAs, both sncRNAs act as important regulators of gene expression, as well as direct contributors to cancer progression. Specifically, more evidence on snoRNAs is needed in the context of the pre-metastatic niche preparation, and vicious cycle occurrence during bone metastasis. Although our knowledge on the causal roles of miRNAs and snoRNAs in cancer is fast expanding, research still needs to be conducted to develop miRNA- and snoRNA-based therapeutics to improve cancer treatment for patients. These molecules can be used as prognostic tools to monitor cancer progression, since they are greatly stable in biological fluids and can reflect physio-pathological changes. However, there are a lack of preclinical trials probably because some of the discovered molecular mechanisms depend on specific contexts or tissues and are failing to be translated in the clinic. The fact that different molecules, such as miRNA and snoRNAs, can cooperate in the same processes could be a clue to better identifying molecular mechanisms specifically associated with cancer and bone metastasis, and thus efficiently targeting them. 

## Figures and Tables

**Figure 1 cancers-15-00242-f001:**
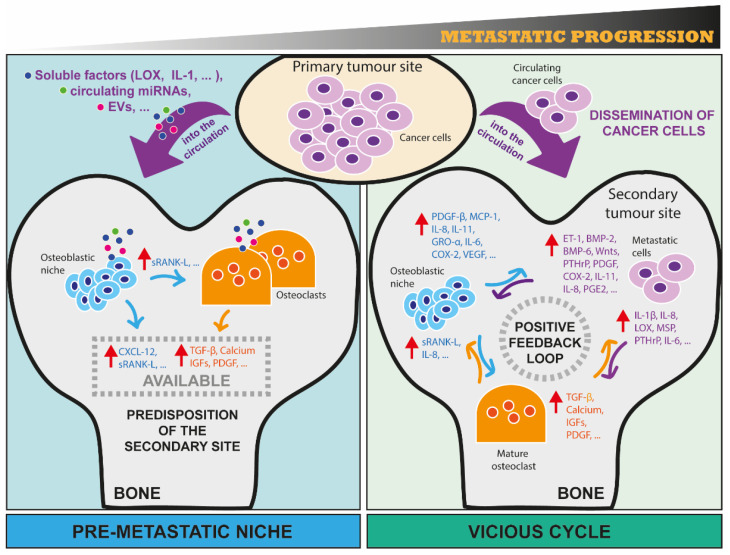
The metastatic progression of cancer cells in bone from the construction of a pre-metastatic niche formation to the positive feedback loop of the vicious cycle. Here, we schematically illustrate two important (and consecutive) processes that occur during bone metastasis. On the left-hand panel, the concept of pre-metastatic niche as one of the first events of metastasis is illustrated. First, cancer cells at the primary tumour site release soluble factors (e.g., LOX, IL1-β), circulating miRNAs, and extracellular vesicles that can reach distant sites, such as bone, through circulation. Once in bone, these factors are taken up by bone resident cells (osteoclasts and osteoblasts), affecting their activities and their secretome, and thus creating an imbalance of bone homeostasis that predisposes this environment to host future metastatic cells. For example, osteoblasts can be induced to secrete CXCL-12 that help metastatic cells to seed in bone, and sRANK-L that can bind to its receptor RANK on osteoclast precursors, promoting their differentiation in mature osteoclasts. Osteoclasts can be induced to increase their bone-resorbing activity, enabling bone matrix-embedded factors (e.g., TGF-β, Calcium, IGFs, PDGF) released from resorbed bone to act on metastatic cells. On the right-hand panel, the concept of vicious cycle between metastatic cancer cells, osteoclasts and osteoblasts is illustrated. First, cells from a primary site physically translocate to bone through circulation. Once in bone, metastatic cells interact with resident cells, taking advantage of bone cell activities to sustain their growth. A positive feedback loop is then established, with metastatic cells producing a few factors that promote osteoclast and/or osteoblast activity, and in turn the activity of bone resident cells is able to further sustain tumour growth through the release of soluble factors, some of them being listed in the figure. Red arrows indicate the increase of production and consequent secretion of the corresponding factors. (EVs: extracellular vesicles; LOX: Lysyl Oxidase; IL-1β/8/6/11: Interleukin 1 Beta/8/6/11; CXCL-12: C-X-C motif chemokine ligand 12; sRANK-L: Soluble receptor activator of nuclear factor kappa-B ligand; TGF-β: Transforming growth factor-beta; IGFs: Insulin-like growth factors; PDGF: Platelet-derived growth factors; ET-1: Endothelin 1; BMP-2/6: Bone morphogenetic protein 2/6; Wnts: Wingless-Type MMTV Integration Site Family; PTHrP: Parathyroid hormone-related protein; MSP: Macrophage-stimulating protein; COX-2: Prostaglandin-endoperoxide synthase 2; PGE2: Prostaglandin E2; MCP-1: Monocyte chemoattractant protein-1; VEGF: Vascular endothelial growth factor).

**Figure 2 cancers-15-00242-f002:**
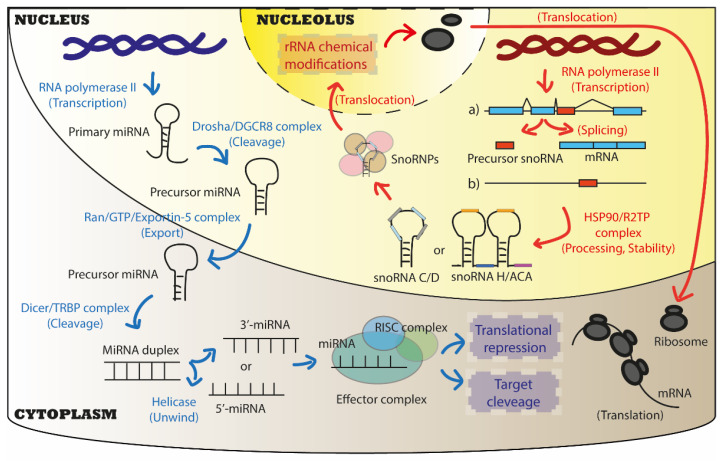
Biogenesis and canonical biological functions of miRNAs and snoRNAs. Both miRNAs and snoRNAs are usually transcribed by the RNA polymerase II (RNA Pol II) in the nucleus of cells. For miRNA biogenesis, two cleavages are requested to obtain a mature miRNA: the first is performed by Drosha/DGCR8 complex in the nucleus, the second by Dicer/TRBP in the cytoplasm. The Ran/GTP/Exportin-5 complex is responsible for the physical translocation of immature miRNAs from the nucleus to the cytoplasm. Finally, one or both strands of the mature miRNA duplex (3′, 5′ filaments) can interact with the RISC complex to form the effector complex that exerts its biological functions in the cytoplasm mainly. Canonical functions of miRNAs are to act as translational repressor by promoting the cleavage of targets or inhibiting their translation. For snoRNA biogenesis, after transcription from genomic intron regions (the most common pathway), or as independent transcripts, snoRNA precursors are processed and stabilised by the HSP90/R2TP complex into mature snoRNAs that usually belong to two main classes: Box C/D and box H/ACA snoRNAs. Once associated with core proteins to form snoRNPs, mature snoRNAs then translocate into a sub-compartment of the nucleus, the nucleolus, where they exert their biological functions. Canonical functions of snoRNAs are to promote chemical modifications on ribosomal RNAs (rRNAs), which directly impacts ribosome function during mRNA translation.

**Figure 3 cancers-15-00242-f003:**
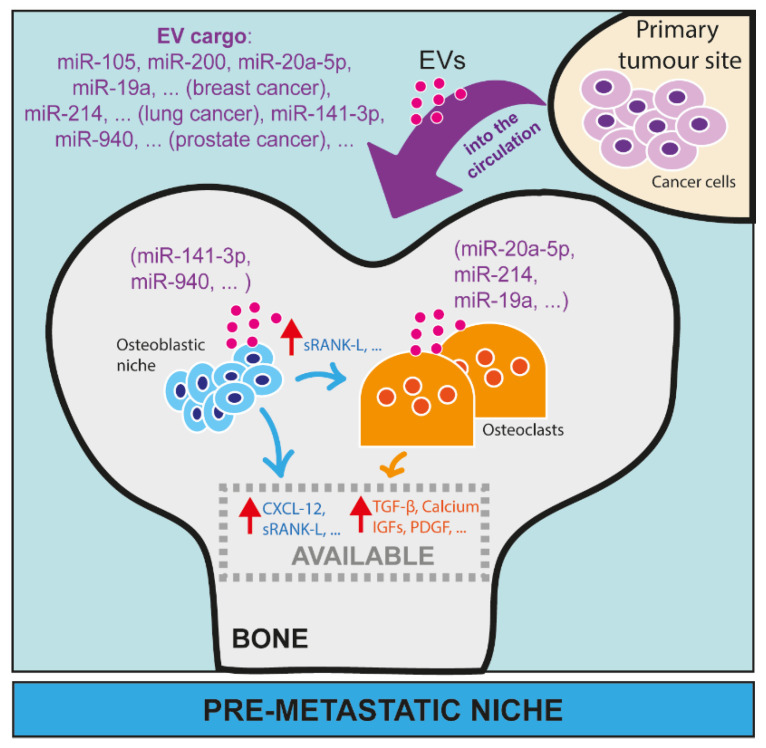
The role of microRNAs in the pre-metastatic niche in bone. MiRNAs have been largely investigated as EV cargo from cancer cells able to remotely affect the activity of cells in distant organs, such as osteoblasts and osteoclasts in bone. Here, we reported some examples of miRNAs from breast cancer (miR-105, miR-200, miR-20a-5p, miR-19a, etc.), lung cancer (miR-214, etc.) and prostate cancer (miR-141-3p, miR-940, etc.) that, as EV cargo, can circulate in blood or lymphatic vessels and reach distant sites, such as bone. In bone, EVs can be taken up by osteoblasts (e.g., miR-141-3p, miR-940) or osteoclasts (e.g*.,* miR-20a-5p, miR-214, miR-19a), and modulate their activity and/or maturation.

**Figure 4 cancers-15-00242-f004:**
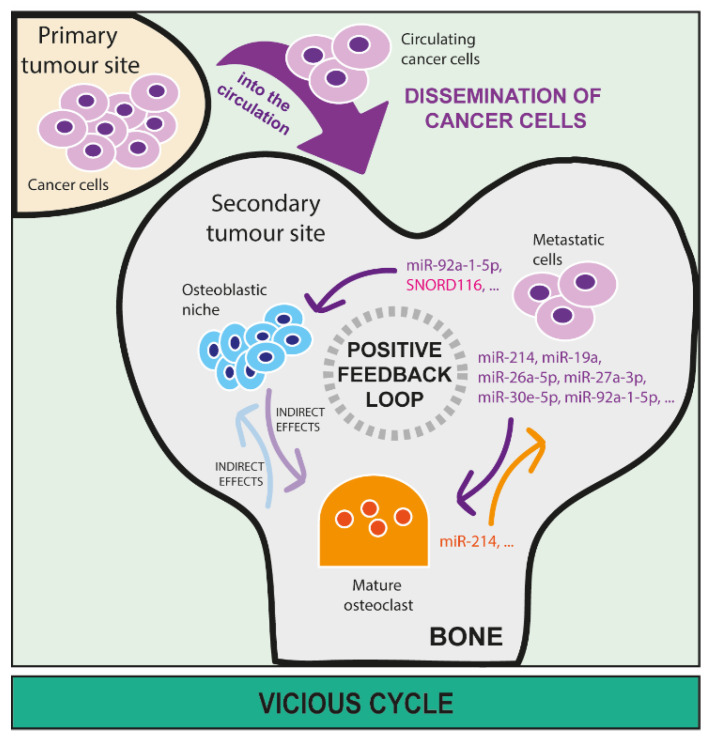
The role of microRNAs and snoRNAs in the vicious cycle in bone. MiRNA and snoRNA release by metastatic cancer cells in bone is directly involved in sustaining a positive feedback loop (also called ‘vicious cycle’) between cancer, osteoclast, and osteoblast cells that further worsen the unbalance of bone homeostasis due to the presence of metastasis. Once metastatic cells disseminate to bone through the blood circulation, they can seed and proliferate in this new micro-environment. Here, cancer-derived miRNAs can directly affect activities of both osteoclasts (miR-214, miR-19a, miR-26a-5p, miR-27a-3p, miR-30e-5p, miR-92a-1-5p, etc.) and osteoblasts (miR-92a-1-5p, SNORD166, etc.) as well as the relationship between osteoclasts and osteoblasts, leading to bone lesion formation. Additionally, miRNAs that derive from osteoclasts (miR-214, etc.) and osteoblasts can directly sustain the growth of metastatic cells in the bone marrow.

**Figure 5 cancers-15-00242-f005:**
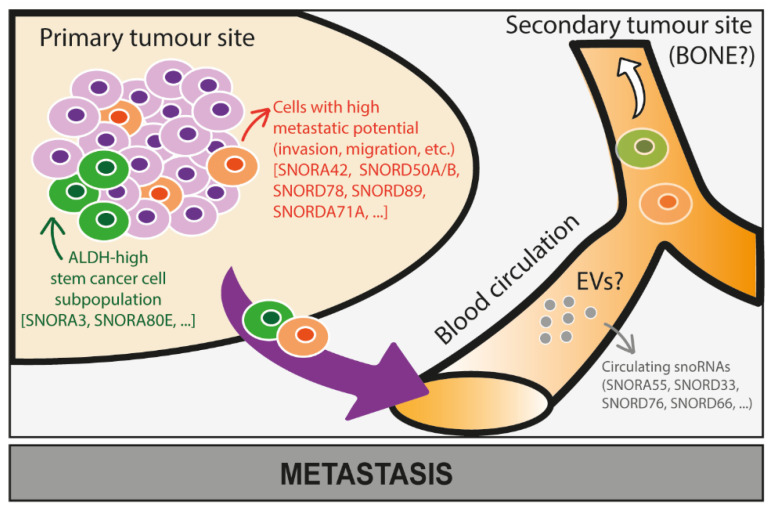
Potential roles of snoRNAs in the metastatic progression of cancer cells. The role of snoRNAs in the metastatic progression of cancer cells in bone is still largely unknown and needs further investigation. Based on some studies conducted on snoRNA expression levels in cancer cells at the primary tumour site, we here speculate that some snoRNAs proven to be associated with stem-like properties (e.g., SNORA3, SNORA80E) and increased metastatic properties—invasion and migration—(e.g., SNORA42, SNORD50A/B, SNORD78, SNORD89, SNORDA71A) of cancer cells might also promote bone metastasis. Moreover, expression levels of snoRNAs (e.g., SNORA55, SNORD33, SNORD76, SNORD66) are dysregulated in circulating cancer cells that might successfully colonise bone, which requires further investigation.

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
