# Peer review of "MiRNAs and snoRNAs in Bone Metastasis: Functional Roles and Clinical Potential"

_cancers, 2022, doi:10.3390/cancers15010242_

Round 1
Reviewer 1 Report
In this manuscript, the authors provide a comprehensive review to highlight the importance of microRNAs (miRNAs) and small nucleolar RNAs (snoRNAs) in metastasis, especially in bone. In the first part of the review, they present the role of miRNAs and snoRNAs in three differentiated functions: the formation of a pre-metastatic niche, maintenance of a tumor-osteoblast and osteoclast vicious cycle in the bone, and then in progression of bone metastasis. Secondly, they discuss the putative role of these noncoding RNAs as diagnostic/prognostic biomarkers and therapeutic targets. The review is written in a direct, educational, and easy-to-follow style. Nevertheless, I would like to comment on some aspects which may benefit from some clarification or specification with the aim to improve the manuscript:
1. Please specify the “snoRNA” acronym at the beginning of the manuscript since this does not come until page 5 (line 193).
· in relation to the latter, please justify the rationale behind specifically reviewing the role of miRNAs and snoRNAs and not other noncoding RNAs
· the names of the different classes of noncoding RNAs may be stated in the introduction to Point 3 (lines 184-201).
2. Within bone, osteoblasts, and osteoclasts are the two main cellular players interacting with tumor cells for the establishment of bone metastasis. Authors may briefly comment on whether it is known if other cellular types (osteocytes, immune cells, fibroblast cells…) may also have a role in bone metastasis.
3. A great part of the review deals with the role of circulating miRNAs and snoRNAs in bone metastasis (either in a free form in biological fluids or as extracellular vesicles [EVs] cargo). However, this is not really addressed until Point 4 (page 8), although EVs are already depicted in Figure 1.
· this should at least be cited in the simple summary and abstract of the manuscript.
· the small round balls in Figures 1 and 3 should be clearly identified as “EVs”. What about Figures 4 and 5?? Please clarify whether information comes from miRNAs and snoRNAs within EVs (depicting vesicles besides ncRNA names) or from free circulating miRNAs and snoRNAs (just ncRNAs names).
· in relation to the latter issue, please review the text in Point 4 and 5 to check whether at any point commented roles have not been attributed to free or EV-embedded miRNAs and snoRNAs.
4. Please check about other putative soluble factors and cytokines involved in the establishment of the pre-metastatic niche. Are VEGF, PGF, IL1β, and LOX the only factors involved in this process?
· please check about correctness for lines 1651- 166: “…osteoblasts can be induced to secrete CXCL-12 and sRANKL, two factors that help metastatic cells to seed in bone”. Maybe it is worth rephrasing this sentence since in the case of sRANKL, this is indirectly mediated by the release of growth factors from the bone matrix after increased osteoclast resorption mediated by sRANKL. This is later explained by the authors.
5. In general, the whole manuscript results a bit lengthy. The authors may try to summarize it while maintaining the essence of the review.
6. Minor considerations:
· please review the use of Greek letters in the figures.
· please check correctness in line 246: “Abnormal miRNA expression in cancer cells are common…”
· please check correctness in line 409: “Whether snoRNA-associated EV might be involved in metastatic progression…”
· please check correctness in line 524-5: “…while SNORA23 overexpression in pancreatic PDAC cell lines promote liver metastases …”
Reviewer 2 Report
The scientific evidence of the involvement os sncRNAs in the development of cancer and metastasis is much higher and stronger for miRNAs than for snoRNAs. This is a consequence of the fact that miRNAs have been under investigation for much longer.
Regarding the use of liquid biopsy for the analysis of circulating sncRNAs, both free and associated with EVs, it is clear that it´s only useful for miRNAs, since snoRNAs are very scarce in serum/plasma. So diagnostic, prognostic (predictive recurrence), and treatment applications in liquid biopsy from circulating sncRNAs are based almost exclusively on miRNAs.
All of the above points to that this Review Article should focus on miRNAs regarding the diagnosis, prognosis, and treatment of tumors through liquid biopsy. This would have important implications for its structure and content (title, abstract, ...). It should include an exclusive section for possible future clinical applications of snoRNAs.
Reviewer 3 Report
The manuscript ID: (2023615) Title: MiRNAs and snoRNAs in Bone Metastasis: Functional Roles and Clinical Potential: I have gone through the whole manuscript and overall, this review article is clear, concise, and well-written. This article is presented for readers to follow the present study rationale. The authors make a systematic contribution to the research literature in this area and summarizes pathophysiology of Bone Metastasis and application of MiRNAs and snoRNA as therapeutic target for bone metastasis. The manuscript is good to publish in a journal like Cancers. But need some necessary changes. Include something more about the earlier work and review of literature.
Major issues:
1. Author needs to discuss more about the limitation and challenges of this study.
2. Did this study provide any impact of MiRNAs and snoRNA as therapeutic regimen on Bone Metastasis in humans? Has there been any attempt to target MiRNAs and snoRNA therapy on Bone Metastasis in clinical trial?
3. The side effect of targeting MiRNAs and snoRNA should be discussed.
Minor issues:
1. Additional references are needed to support the statement from line 252 to 266.
2. The manuscript requires careful attention for possible grammatical errors.
Round 2
Reviewer 2 Report
I believe that the role attributed to snoRNAs has been improved, according to scientific evidence, in bone metastasis, giving more importance to miRNAs in this process. Also it is important to stablish that EV-encapsulated snoRNAs is the only way of traveling these molecules in the bloodstream.